# Chitin Deacetylase, a Novel Target for the Design of Agricultural Fungicides

**DOI:** 10.3390/jof7121009

**Published:** 2021-11-25

**Authors:** Jesús M. Martínez-Cruz, Álvaro Polonio, Riccardo Zanni, Diego Romero, Jorge Gálvez, Dolores Fernández-Ortuño, Alejandro Pérez-García

**Affiliations:** 1Departamento de Microbiología, Facultad de Ciencias, Universidad de Málaga, 29071 Málaga, Spain; jesusmcruz@uma.es (J.M.M.-C.); polonio@uma.es (Á.P.); diego_romero@uma.es (D.R.); aperez@uma.es (A.P.-G.); 2Instituto de Hortofruticultura Subtropical y Mediterránea “La Mayora”, Universidad de Málaga, Consejo Superior de Investigaciones Científicas (IHSM-UMA-CSIC), 29010 Málaga, Spain; 3Molecular Topology and Drug Design Unit, Department of Physical Chemistry, University of Valencia, 46010 Valencia, Spain; riccardo.zanni@uv.es (R.Z.); jorge.galvez@uv.es (J.G.)

**Keywords:** chitin, chitin deacetylase, chitin-triggered immunity, EDTA, *Podosphaera xanthii*, powdery mildews, RNAi silencing

## Abstract

Fungicide resistance is a serious problem for agriculture. This is particularly apparent in the case of powdery mildew fungi. Therefore, there is an urgent need to develop new agrochemicals. Chitin is a well-known elicitor of plant immunity, and fungal pathogens have evolved strategies to overcome its detection. Among these strategies, chitin deacetylase (CDA) is responsible for modifying immunogenic chitooligomers and hydrolysing the acetamido group in the N-acetylglucosamine units to avoid recognition. In this work, we tested the hypothesis that CDA can be an appropriate target for antifungals using the cucurbit powdery mildew pathogen *Podosphaera xanthii*. According to our hypothesis, RNAi silencing of *PxCDA* resulted in a dramatic reduction in fungal growth that was linked to a rapid elicitation of chitin-triggered immunity. Similar results were obtained with treatments with carboxylic acids such as EDTA, a well-known CDA inhibitor. The disease-suppression activity of EDTA was not associated with its chelating activity since other chelating agents did not suppress disease. The binding of EDTA to CDA was confirmed by molecular docking studies. Furthermore, EDTA also suppressed green and grey mould-causing pathogens applied to oranges and strawberries, respectively. Our results conclusively show that CDA is a promising target for control of phytopathogenic fungi and that EDTA could be a starting point for fungicide design.

## 1. Introduction

Chemical control practices have been critical in preventing losses due to plant diseases, especially fungal diseases. Unfortunately, fungicides marketed for the control of diseases on food crops are under pressure. One of the best examples of fungal diseases in which chemicals play a key role in disease management is powdery mildews. Among the economically important plants they affect are cereals, grapes, and many vegetables and ornamental plants [1]. Immense expenditures are made annually for fungicides to control powdery mildews; therefore, they are considered “strategic diseases” by the agrochemical industry. Unfortunately, the problem of fungicide resistance is of paramount importance in powdery mildew fungi. One example is the cucurbit powdery mildew pathogen *Podosphaera xanthii* [2]. In southern Spain, resistance to the most popular anti-powdery mildew fungicides has been reported [3,4,5], with multiresistant isolates found in the areas of more intense cropping [5]. Therefore, to be able to maintain chemical diversity for disease management, there is a pressing demand to identify and develop new agrochemicals.

Fungal cell walls are dynamic structures that are essential for cell viability, morphogenesis, and pathogenesis, and they are the first defence barrier against fungal pathogens [6]. During plant colonization, the fungal cell wall is exposed to the action of plant degradative enzymes such as chitinases and β-glucanases [7,8]. Chitin is an important structural component of fungal cell walls and a well-known elicitor of plant immunity. As a consequence of plant enzymatic activities, small chitin oligomers are released that can be recognized by plant receptors such as CERK1, promoting the activation of the chitin-specific signalling cascade [9,10]. In response, fungal pathogens have developed strategies to overcome chitin detection, including alterations in the composition of cell walls by modifying their carbohydrate chains and the secretion of effectors to provide protection to the cell wall or to bind immunogenic oligomers, thus preventing the activation of host immune responses [11].

One of these strategies is the conversion of cell wall chitin into chitosan by chitin deacetylase (CDA). This enzyme is a conserved protein in fungi that catalyses the hydrolysis of the N-acetamido group in the N-acetylglucosamine units of chitin to convert it to chitosan, the deacetylated derivative of chitin, a poor substrate for chitinases and a compound with elicitor activity notably lower than chitin [11,12]. CDA is a dispensable protein for in vitro growth. For example, yeast *cda* mutants were viable under standard laboratory conditions; however, they showed increased sensitivity to hydrolytic enzymes [13]. Regarding plant pathogens, *cda* mutants of the soil fungal pathogens *Verticillium dahliae* and *Fusarium oxysporum* showed normal colony morphology, but interestingly, they exhibited markedly reduced virulence in cotton plants and no longer induced wilt symptoms [14]. Similarly, virulence defects have also been observed in *Ustilago maydis cda* mutants [15]. Given the wide conservation of CDA in fungi, the deacetylation of chitin oligomers to avoid host perception by chitin receptors is anticipated to be a common and conserved strategy of plant pathogenic fungi for survival in a host [14].

As mentioned above, powdery mildew fungi (*Erysiphales*) are probably the largest group of fungal pathogens in which the problem of fungicide resistance is of greatest concern, with hundreds of reports describing control failures and resistance to many chemical classes in powdery mildew species that affect major crops [16]. To address this problem, in our laboratory, we used functional genomics approaches to identify essential proteins for powdery mildew pathogenesis, employing the cucurbit pathogen *P. xanthii* as a model system [17]. Using RNAi silencing, we demonstrate that CDA is a key protein for powdery mildew virulence. In addition, we also found that treatments with EDTA, a well-known CDA inhibitor, suppresses powdery mildew disease by activating chitin signalling. Our results suggest that CDA could be an interesting target for the design of fungicides against powdery mildews and other plant pathogenic fungi and that the EDTA molecule could be a lead fungicide.

## 2. Materials and Methods

### 2.1. Plants, Fungi, Bacteria and Culture Conditions

Zucchini (*Cucurbita pepo* L.) cv. Negro Belleza (Semillas Fitó, Barcelona, Spain) and melon (*Cucumis melo* L.) cv. Rochet (Semillas Fitó) plants were used for *P. xanthii* growth and RNAi silencing experiments. Plants were cultivated in a growth chamber at 24 °C under a 16 h light/8 h dark cycle. For the growth of the *P. xanthii* isolate 2086, disinfected cotyledons of zucchini maintained in 8 cm Petri dishes with Bertrand medium (sucrose 40 g L^−1^, agar 10 g L^−1^, benzimidazole 30 mg L^−1^, pH 7.0) were used as previously described [18]. The postharvest fungal pathogens *Botrytis cinerea* and *Penicillium digitatum* used in fruit assays were maintained on CZAPEK DOX agar (sodium nitrate 2 g L^−1^, potassium chloride 0.5 g L^−1^, magnesium glycerophosphate 0.5 g L^−1^, ferrous sulphate 0.01 g L^−1^, potassium sulphate 0.35 g L^−1^, sucrose 30 g L^−1^, agar 12 g L^−1^, pH 6.8) and potato dextrose agar (PDA) (potato extract 4 g L^−1^, glucose 20 g L^−1^, agar 15 g L^−1^, pH 5.6) and PDA media at 22 °C and 25 °C, respectively. For agroinfiltration and RNAi silencing, *Agrobacterium tumefaciens* C58C1 was used and grown at 28 °C in lysogeny broth (LB; “Lab-Lemco” powder 1 g L^−1^, yeast extract 2 g L^−1^, peptone 5 g L^−1^, sodium chloride 5 g L^−1^, pH 7.4) medium with rifampicin (50 µg mL^−1^) and tetracycline (5 μg mL^−1^). This medium was modified by adding the corresponding selective marker for each binary vector. For the maintenance, construction, and propagation of RNAi silencing vectors, the *Escherichia coli* strain DH5α was used. The *E. coli* strains was grown at 37 °C in LB medium with the corresponding antibiotic [17].

### 2.2. Sequence Analysis

In a previous study on the *P. xanthii* epiphytic transcriptome and the corresponding deduced secretome [19], a putative chitin deacetylase (CDA) transcript was identified. The search for homologous amino acid sequences in databases was carried out using the BLAST tool (Basic Local Alignment Search Tool) from NCBI (ncbi.nlm.nih.gov, accessed on 22 January 2020). The signal peptide was identified using the SignalP 4.1 server [20]. The sequences were aligned using the UniProt web server (UniProt.org, accessed on 23 January 2020). A search for conserved domains was inspected manually.

### 2.3. RNA Isolation and cDNA Synthesis

For RNA isolation, *P. xanthii*-infected melon cotyledons were finely ground in a mortar with liquid nitrogen and pestle. Subsequently, total RNA was isolated using the TRI Reagent^®^ RNA isolation system (Sigma–Aldrich, St Louis, MO, USA) according to the manufacturer’s recommendation. Contaminating DNA was removed using a TURBO DNA-free kit^®^ (Invitrogen, Carlsbad, CA, USA). The total RNA concentration was estimated using a NanoDrop spectrophotometer ND-1000^®^ (Thermo Fisher Scientific, Waltham, MA, USA). cDNA was synthesized using Superscript^®^ III Reverse Transcriptase (Invitrogen) with oligo dT(20) primers (Invitrogen) according to the manufacturer’s recommendation.

### 2.4. Plasmid Construction

For RNAi silencing of the *P. xanthii* CDA gene (*PxCDA*) and the melon chitin kinase receptor gene (*CmCERK1*), the vector pB7GWIWG2(II) [21] and Gateway cloning technology (Invitrogen) were used essentially as previously described [17]. Specific primers with attB1 or attB2 tails (Appendix A) were used to amplify fragments of 842 and 614 bp of *PxCDA* and *CmCERK1* (XP_008455461.1), respectively, from the cDNA previously obtained (see above). The resulting plasmids pPxCDA-HIGS and pCmCERK1-HIGS were checked by PCR and sequencing. These plasmids were propagated and maintained in *E. coli* strain DH5α. For the RNAi silencing experiments, these plasmids were introduced into *A. tumefaciens* strain C58C1 by electroporation and were verified by PCR and restriction enzyme digestion.

### 2.5. A. Tumefaciens-Mediated Host-Induced Gene Silencing (ATM−HIGS)

To determine the role of *PxCDA* in *P. xanthii* development, the ATM-HIGS assay was conducted as previously described [17]. For this assay, the RNAi silencing plasmids pPxCDA-HIGS and CmCERK1-HIGS, as well as the plasmid pB7GWIWG2(II) (empty vector, negative control), were used. Briefly, transformed *A. tumefaciens* cells were induced with 200 µM acetosyringone and grown in 5 mL of LB medium supplemented with rifampicin (50 µg∙mL^−1^) and spectinomycin (100 µg∙mL^−1^) or kanamycin (50 µg∙mL^−1^) at 28 °C and 200 rpm in an orbital shaker overnight. Subsequently, the different *Agrobacterium* cultures corresponding to the different RNAi silencing constructs were washed twice in washing buffer and 200 µM acetosyringone. To induce the Vir proteins, the bacterial cells were incubated for 2 h at 28 °C in the same washing buffer without agitation. Then, the *Agrobacterium* cells were centrifuged for 10 min at 4000 rpm and 28 °C and resuspended in MES buffer, and their optical density at 600 nm was adjusted to 0.5–1.0 in MES buffer. Finally, agroinfiltration into the abaxial surface of melon cotyledons with the different cell suspensions was carried out using 1-mL syringes without the needle. For cosilencing experiments, equal volumes of *Agrobacterium* cell suspensions carrying the pPxCDA-HIGS and pCmCERK1-HIGS silencing constructs were mixed before agroinfiltration. The agroinfiltrated cotyledons were maintained in a growth chamber under a 16 h light/8 h dark cycle at 24 °C for 24 h until inoculation with fresh *P. xanthii* conidial suspensions (1 × 10^5^ conidia mL^−1^) by pulverization. Then, the cotyledons were maintained under the same conditions until analysis.

### 2.6. RT-qPCR

The in planta expression analysis of HIGS constructs, as well as the expression analysis of the fungal *PxCDA* and melon *CmCERK1* genes, were carried out by RT–qPCR. The primers used for these analyses (Appendix A) were designed using Primer3 software [22,23]. The *P. xanthii* β-tubulin gene *PxTUB2* (KC333362) and the *C. melo* actin-7 gene *CmACT7* (XM_008462689.2) were used as normalization reference genes [24,25]. The effects of gene silencing on the expression of *P. xanthii* and melon genes during HIGS assays were analysed using cDNA obtained from agroinfiltrated and *P. xanthii*-infected melon cotyledons at 24 h after inoculation. RT–qPCR was carried out in a CFX384 Touch Real-Time PCR detection system (Bio–Rad, Hercules, CA, USA) using SsoFast EvaGreen Supermix (Bio–Rad) according to the manufacturer’s instructions with the following cycling conditions: enzyme activation step at 95 °C for 30 s, followed by 40 cycles at 95 °C for 5 s, and 65 °C for 5 s. All reactions were performed in quadruplicate. After amplification, the data were processed by CFX Manager Software (Bio–Rad), and the amplicon sizes were confirmed by visualization on 2% agarose gels.

### 2.7. Hastorial Counts and Production of Hydrogen Peroxide

The progress of *P. xanthii* growth and the in situ accumulation of hydrogen peroxide (H_2_O_2_) in plant epidermal cells were determined by histochemical analysis according to the DAB uptake method described previously [17]. Briefly, cotyledon disks were incubated in 0.1% DAB (3,3′-diaminobenzidine) (pH 3.8) (Sigma–Aldrich, Darmstadt, Germany) overnight in the dark and at room temperature. After incubation, disks were immersed in boiling ethanol to stop the reaction and to be cleared. Finally, the cotyledon disks were analysed under a light microscope, and brownish-red precipitates corresponding to H_2_O_2_ accumulation were observed. The fungal penetration points were visualized as dark-brown spots spread along the *P. xanthii* hyphae.

### 2.8. Leaf Disc and Plant Assays

To test the capacity of the inhibition of selected compounds, a leaf disc assay was used as previously described [3], with minor modifications. Briefly, 1 cm diameter discs were obtained from zucchini cotyledons and placed in 8 cm Petri dishes containing Bertrand medium. The plates were incubated under a 16 h light/8 h dark cycle at 25 °C for 24 h for disc acclimatization. After this period, the discs were inoculated with a 5 μL drop of a conidial suspension (1 × 10^5^ conidia mL^−1^) deposited at the centre of the discs and incubated under the same conditions for 24 h. After this incubation, the discs were immersed in the corresponding compound solution (acetic acid and lactic acid, 50 and 100 mM; TAE (Tris-acetate-EDTA) buffer, 1× and 5×; EDTA (ethylenediaminetetraacetic acid) 0.5, 2, 10, 20 and 50 mM; 8 quinolinol, TEMED (N,N,N’,N’-tetramethylethylenediamine) and trisodium citrate, 10 and 20 mM; ferrous sulphate, 1, 10 and 20 mM) and placed back in Petri dishes. Finally, the discs were incubated for 6–12 days until data collection.

For plant assays, 4-week-old melon plants were used. For these assays, leaves were inoculated by spreading with the conidial suspension (10^5^ conidia mL^−1^). Twenty-four hours after inoculation, leaves were sprayed with the compound solution (20 mM EDTA). Plants were then maintained in a green chamber under a 16 h light/8 h dark cycle at 25 °C until sampling.

### 2.9. Molecular Docking

The crystal structure of *Aspergillus nidulans* chitin deacetylase (2y8u) [26], was obtained from the RCSB Protein Data Bank (https://www.rcsb.org, accessed on 14 September 2020) and used to carry out docking analysis using the command line of Autodock Vina software [27], with 9 modes. The grid box (center_x = −2.99; center_y = −39.534; center_z = 27.103; size_x = 12.75; size_y = 12.75; size_z = 15) was calculated according to catalytic and metal binding residues described by Liu et al. [26]. These coordinates were used to predict the interactions of chitotriose (ChemSpider ID 108813) and EDTA (ChemSpider ID 5826) ligands with the chitin deacetylase from *A. nidulans*. The best results in terms of affinity (kcal/mol) were visualized using Autodock Tools [28].

### 2.10. Fruit Assays

To analyse the effects of EDTA on other plant diseases, fruit assays using strawberries and oranges were performed to test the inhibition of the necrotrophic fungal pathogens *Botrytis cinerea* and *Penicillium digitatum*, respectively. For the strawberry assay, commercially grown strawberry fruits were rinsed three times with sterile water for 30 s each and allowed to air dry. For *B. cinerea* inoculation, fruits were punctured at one point using a sterile syringe tip. Ten microlitres of conidial suspension (1.7 × 10^6^ conidia mL^−1^) was inoculated into the wounds by injection using the same syringe. The fruits were maintained in plastic boxes and incubated at 22 °C [29]. Eight hours after inoculation, EDTA treatments were applied by immersing the strawberries in EDTA solutions (20, 50 and 100 mM). Finally, the fruits were maintained in boxes at 22 °C in the dark. During the first 24 h, the plastic boxes were sealed with parafilm to maintain a high relative humidity.

For the orange assay, the procedure was similar to that used for strawberries but with some modifications. To infect the oranges, 30 μL of conidial suspension (1.2 × 10^6^ conidia mL^−1^) was deposited on the fruit surface, which was punctured by pricking with a sterile syringe tip [30]. The fruits were maintained in plastic boxes and incubated at 25 °C in the dark. Eight hours after inoculation, EDTA treatments were applied by immersing the oranges in EDTA solutions for 2 min. Finally, the fruits were maintained under the same conditions described previously.

### 2.11. Statistical Analysis

When required, statistical analysis of data was carried out by IBM SPSS v. 25 software (SPSS, Chicago, IL, USA) using Fisher’s least significant difference test (LSD).

## 3. Results

### 3.1. Chitin Deacetylase Is an Essential Protein for P. xanthii Development

Given the important role of chitin-triggered immunity in plant basal defence, in this work, we aimed to determine the role of chitin deacetylase in powdery mildew virulence using the pathosystem *P. xanthii*-melon. A sequence of a putative chitin deacetylase transcript (*PxCDA1*) was previously identified in the epiphytic transcriptome of *P. xanthi*. BLAST analysis found 97% query coverage and 66% identity with a chitin deacetylase of *Blumeria graminis* f. sp. *hordei* DH14 (CCU76842) [31]. Using specific primers, a fragment of 1.3 kb containing the putative *P. xanthii CDA* gene was amplified from genomic DNA. The analysis of both sequences revealed that the *PxCDA* gene was composed of six exons and five introns, all of which showed the consensus GT-AG sequence of fungal introns [32]. This transcript sequence was deposited in GenBank as KX495502. Later, a second *P. xanthii* CDA transcript was found that was 30 nucleotides shorter. The sequence of this second transcript, designated as *PxCDA2*, was also deposited in Genbank (KX495503). In addition, the *P. xanthii* genome [33], was scanned for the presence of CDA genes, resulting in the identification of only one CDA gene. The deduced *P. xanthii* CDA proteins have the typical motifs of chitin deacetylases (Figure 1). As shown in the figure, the residues involved in the binding of iron and those that make up the catalytic site are perfectly conserved among several CDA of various fungi.

To determine the role of CDA in the interaction, RNAi silencing experiments were conducted using an RNAi silencing vector designed to knock out the expression of both *P. xanthii* CDA transcripts. After checking the correct expression in planta of the silencing construct (Appendix A), the expression of the CDA transcripts was reduced by 50% (Appendix A), which led to a dramatic reduction in fungal growth that was coupled to a strong production of hydrogen peroxide (Figure 2), suggesting the presumably rapid activation of chitin-triggered immunity. To link the observed responses with the activation of this plant immunity, RNAi silencing experiments were performed to simultaneously silence the *PxCDA* gene and the plant chitin elicitor receptor kinase gene *CmCERK1*. As anticipated, in co-silenced tissues, fungal growth was fully restored, and the production of hydrogen peroxide was considerably reduced (Figure 2), confirming that the response activated in the plant after the silencing of *PxCDA* was indeed chitin-triggered immunity.

### 3.2. Small Carboxylic Acids Also Suppress Powdery Mildew Disease

Once the fundamental role of chitin deacetylases in *P. xanthii* development was demonstrated, we next asked whether the application of chitin deacetylase inhibitors would also have a negative impact on disease development. For that, we tested different concentrations of small carboxylic acids, known inhibitors of chitin deacetylase in vitro [34], such as acetic acid, lactic acid, or combinations such as TAE (Tris-acetate-EDTA) buffer, using a leaf disc assay (Appendix A). The inhibition of fungal growth was observed in all cases, and the most remarkable effect was that of TAE buffer (Figure 3A). Since the effect of acetic acid was not very pronounced, we studied in detail the effect of EDTA (ethylenediaminetetraacetic acid), the other carboxylic acid present in TAE buffer. Compared with the other compounds, the effect of EDTA was dramatic at 20 mM, causing 100% inhibition of fungal growth (Figure 3B).

Given that EDTA is well known as a chelating agent, we wanted to investigate the effects of other chelating agents on disease-suppression activity, since metal ions seem to be necessary for chitin deacetylase activity [12,35,36]. Chelating agents such as 8-quinolinol, TEMED (N,N,N’,N’-tetramethylethylenediamine), or trisodium citrate had no effect on disease development (Appendix A and Figure 4A). Similarly, metal ions such as iron (II) did not affect the efficacy of EDTA in suppressing powdery mildew disease (Figure 4B). Therefore, we concluded that the EDTA efficacy on disease control is unlikely to be due to its chelating effect.

### 3.3. EDTA Inhibits P. xanthii Development by Targeting CDA and Eliciting Chitin-Triggered Immunity

To determine whether the disease-suppressive effect of EDTA was due to the inhibition of chitin deacetylase and the subsequent activation of chitin-triggered immunity, EDTA treatments were applied to melon plants in which the chitin receptor gene *CmCERK1* had been previously silenced. As shown in Figure 5, the effects of EDTA on *P. xanthii* growth and production of hydrogen peroxide by plant cells were very similar to those observed after *PxCDA* silencing. However, these effects of EDTA were partially restored in plants in which *CmCERK1* expression had been knocked down. As previously observed for the silencing of *PxCDA*, these results are consistent with the activation of chitin-triggered immunity after putative inhibition of CDA by EDTA.

To analyse the interactions of CDA protein with ligand (chitin) and inhibitor (EDTA), molecular docking analyses were performed, using for these experiments the 3D model of a CDA protein of *A. nidulans* (2Y8U) [26] and the 3D models of chitotriose, the chitin trimer, and EDTA. First, we conducted the molecular docking of chitotriose and AnCDA to identify the catalytic pocket. As expected, the analysis revealed the previously mapped catalytic site [26], with the residues Tyr138 and His196 involved in the binding to chitotriose (Table 1, Figure 6). After the identification of the catalytic site, a similar analysis was conducted with the EDTA molecule. The molecular docking of EDTA to the AnCDA protein model revealed the binding of EDTA to the enzyme catalytic pocket, showing four residues involved in this interaction: Tyr138 (previously identified in chitin binding), His97, Tyr166, and His199 (Table 1, Figure 6). These results provided additional evidence that the CDA enzyme was a target of EDTA. Collectively, these results suggested that the mechanism of action of the suppression of fungal growth by EDTA was the inhibition of the CDA enzyme and the subsequent activation of chitin signalling by the recognition of nondeacetylated chitin oligomers by plant receptors.

### 3.4. EDTA Treatments Also Suppress Other Fungal Diseases

Given the efficacy of EDTA in controlling cucurbit powdery mildew disease, we wanted to determine whether EDTA was able to control other fungal diseases, especially diseases very different from powdery mildews such as those produced by necrotrophic fungi such as *Botrytis cinerea* (grey mould) or *Penicillium digitatum* (green mould). Based on the results previously obtained in melon leaf disc and plant assays, curative treatments with EDTA were conducted on strawberry and orange fruits previously inoculated with the corresponding fungal pathogens *B. cinerea* and *P. digitatum*, respectively. As shown in Figure 7, EDTA treatments were able to reduce and suppress grey and green mould symptoms, although at concentrations higher than those used to suppress cucurbit powdery mildew (20 mM), 50 mM and 100 mM EDTA, respectively. It is interesting to note that, for the case of strawberry grey mould, full protection was achieved at 50 mM EDTA; however, the fungus was able to infect fruit treated with 100 mM EDTA. This is probably due to the phytotoxic effect of EDTA at this high concentration that hampered the activation of host responses, thus allowing fungal growth on damaged tissue.

## 4. Discussion

Powdery mildews are diseases caused by biotrophic ascomycete fungi from the order *Erysiphales* [1]. Due to their peculiar lifestyle as biotrophs, many aspects of their biology, especially those related to interactions with their hosts, remain unknown [17]. One of the most important aspects of these interactions is related to the ability of these pathogens to suppress the activation of so-called pathogen-associated molecular pattern (PAMP)-triggered immunity, such as chitin-triggered immunity [37]. Regarding powdery mildew fungi, two mechanisms have recently been described: suppressors of chitin signalling in *P. xanthii*, the secretion of EWCA (effectors with chitinase activity) effectors at pathogen penetration sites, and the activity of a haustorial-expressed LPMO (lytic polysaccharide monooxygenase). In both cases, the molecular strategy is very similar; essentially, these proteins break down immunogenic chitin oligomers into smaller and non-immunogenic molecules, thus circumventing chitin signalling [38,39].

Another mechanism involved in the battle for chitin recognition is the activity of the enzyme chitin deacetylase (CDA). Since CDA is a conserved protein in fungi, in this work, we aimed to determine the role of CDA in powdery mildew virulence. Using an RNAi silencing approach [17], we concluded that CDA is a key protein for powdery mildew virulence, since the silencing of the *P. xanthii CDA* gene led to the suppression of fungal growth and to a rapid activation of plant defence responses dependent on the CERK1 plant receptor, and the co-silencing of fungal CDA and plant chitin receptor genes restored fungal growth. These findings suggested that the suppression of fungal development observed upon silencing the fungal *CDA* gene was a consequence of the activation of chitin-triggered immunity [9]. These results supported the idea that CDA also plays an important role in powdery mildew virulence and suggested that a novel strategy for powdery mildew control could be the interference of mechanisms of suppression of chitin-triggered immunity, such as CDA [11].

Due to their epiphytic parasitic nature, powdery mildews are easy targets for fungicides. In fact, chemical control is the main control tool of most powdery mildew diseases [40]. Once the important role played by CDA in disease establishment had been demonstrated, we conducted a search for molecules that could inhibit its activity under the premise that inhibition of CDA activity should also result in the suppression of fungal growth. In previous works on CDA from *Colletotrichum* sp. and other fungal species, it was observed that the presence of certain carboxylic acids, especially EDTA, causes a decrease in enzyme activity in vitro [32,34,41,42,43]. In addition, as suggested by other authors, the negative effect of carboxylic acids could be due to an effect of negative feedback caused by acetate, the carboxylic acid precursor [43]. To test the hypothesis that the inhibition of CDA could also lead to the suppression of powdery mildew growth, assays were conducted using melon discs. The results found that carboxylic acids, especially EDTA, could also arrest fungal growth. Interestingly, EDTA treatments activated a plant defence response similar to that obtained with *PxCDA* silencing. To investigate whether this host response could be related to chitin signalling, experiments were conducted using *CERK1*-silenced plants and treated with EDTA. As observed in plants in which fungal CDA and plant chitin receptors were co-silenced, fungal growth was partially but significantly restored, supporting our hypothesis that the inhibition of CDA activity by EDTA may lead to the activation of chitin signalling by interference with the chitin deacetylation process [14].

Blast analyses showed that CDA genes are widely present in fungi and that different active domains are also conserved. Based on this observation, we hypothesized that it should also have inhibitory activity against other phytopathogenic fungi. This hypothesis was validated using two additional pathosystems, strawberry—*B. cinerea* and orange—*P. digitatum*—demonstrating that EDTA could also arrest diseases caused by necrotrophic fungi and illustrating that CDA could be a nice target not only for powdery mildew fungi but also for other fungal plant pathogens. At this point, one might argue that the antifungal activity of EDTA could also be related to its chelating activity. However, the use of other chelating agents or the use of EDTA in the presence of metal ions did not affect its anti-powdery mildew activity. From these observations, we concluded that the chelating activity of EDTA was not responsible for fungal growth inhibition in these experiments. Furthermore, molecular docking experiments showed that the most likely site of binding of EDTA to the CDA protein is the chitin deacetylase domain in the enzyme catalytic pocket, suggesting that the inhibitory role of EDTA could be associated with its ability to interact with the CDA protein, as demonstrated for other carboxylic acids [34,40,41]. These results supported a direct interaction between the EDTA molecule and CDA protein and suggested the role of EDTA as a CDA inhibitor.

The concentrations of EDTA that were considered fungicidal in this work are relatively high for practical use. However, our results clearly show that chitin deacetylase could be a promising target for fungicide design. The fungal cell wall is an interesting target for fungicides. In fact, chitin synthesis is considered a safe target for fungicides, as chitin is present in fungi (and arthropod exoskeletons) but absent in plants and mammals [6]. Currently, there is only one commercial fungicide targeting chitin biosynthesis, a peptidyl pyrimidine nucleoside called polyoxin D, a chitin synthase inhibitor [44]. Therefore, the inhibition of chitin deacetylase would be a new mode of action and a new tool to control powdery mildews and other phytopathogenic fungi. In this sense, the EDTA molecule could be an interesting lead chemical for fungicide design. One of the most promising computer-aided drug design methods is molecular topology [45]. Unlike the rest of the quantitative structure–activity relationship (QSAR) methods, the molecular topology paradigm uses only pure mathematical descriptors. Defined as a part of mathematical chemistry, molecular topology is basically related to the assimilation between molecules and graphs so that it can depict molecular structures through graph theoretical indices. By this approach, excellent results have been obtained in the design and selection of new drugs in different medical fields [46]. Based on the results presented in this work, we are currently working on the design of chitin deacetylase inhibitors using molecular topology approaches.

## Figures and Tables

**Figure 1 jof-07-01009-f001:**
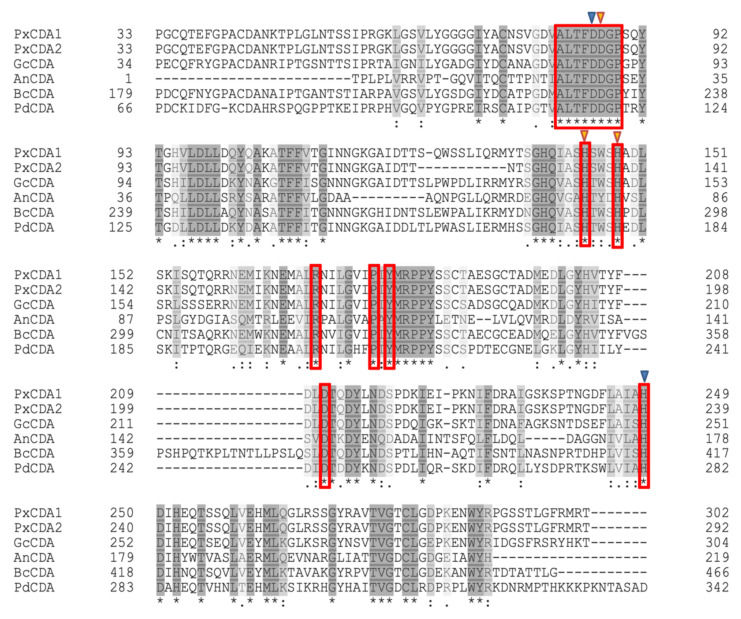
Amino acid sequence alignment of mature chitin deacetylases from *P. xanthii* and chitin deacetylases from various fungi. PxCDA1 (*P. xanthii*, KX495502), PxCDA2 (*P. xanthii*, KX495503), GcCDA (*Golovynomices cichoracearum*, RKF73335), AnCDA (*A. nidulans*, EAA66447), BcCDA (*B.*
*cinerea*, CCD52127), and PcCDA (*P.*
*digitatum*, XP_014535670). The amino acid similarity is shown in grey, with the darkest grey amino acids being the most similar. Conserved amino acids are marked with an asterisk. Conserved residues that are important for enzyme activity are shown in red boxes. The orange arrows show the amino acids involved in iron binding. The blue arrows show the residues that are part of the catalytic site. The sequences were aligned using UniProt web server.

**Figure 2 jof-07-01009-f002:**
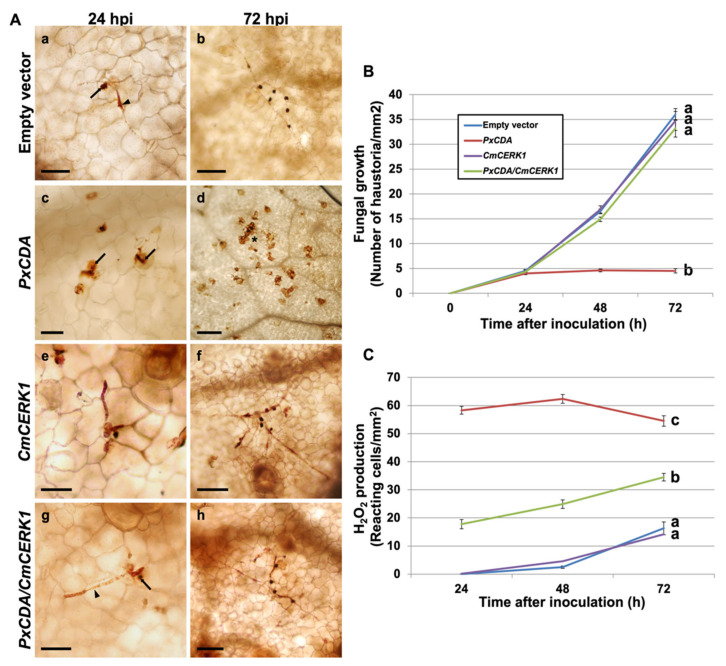
Effect of silencing *PxCDA* on fungal growth and production of hydrogen peroxide. RNA interference (RNAi) silencing was carried out by *A. tumefaciens*-mediated host-induced gene silencing. An empty vector was used as a negative control. Silencing of the melon chitin receptor gene *CmCERK1* was carried out to interfere with host chitin signalling. (**A**) Visualization of fungal structures and H_2_O_2_-producing cells. The detection of hydrogen peroxide was performed by the 3,3′-diaminobenzidine (DAB) uptake method. Pictures were taken at 24 and 72 h after inoculation with *P. xanthii*. Arrowheads indicate *P. xanthii* haustoria; arrows indicate *P. xanthii* hyphae, and asterisks indicate reactive epidermal cells with H_2_O_2_ accumulation. Bars = 50 μm (**a**,**c**,**e**,**g**), 100 μm (**b**,**f**,**h**), and 200 μm (**d**). (**B**) Estimation of fungal growth by haustorial counting. The growth of *P. xanthii* is expressed as the number of haustoria per mm^2^ of transformed tissue. The values are the means of 30 samples from three independent experiments ± standard error. (**C**) Time course analysis of the accumulation of H_2_O_2_ in melon cotyledons. The detection of H_2_O_2_ was performed by the DAB uptake method. The reactive epidermal cells were identified as those containing brown–red precipitates (asterisks). Data represent the number of reactive cells per mm^2^ and are the means of 30 samples from three independent experiments ± standard error. Values with different letters in (**B**,**C**) are significantly different at *p* = 0.05 according to Fisher’s least significant difference test (LSD).

**Figure 3 jof-07-01009-f003:**
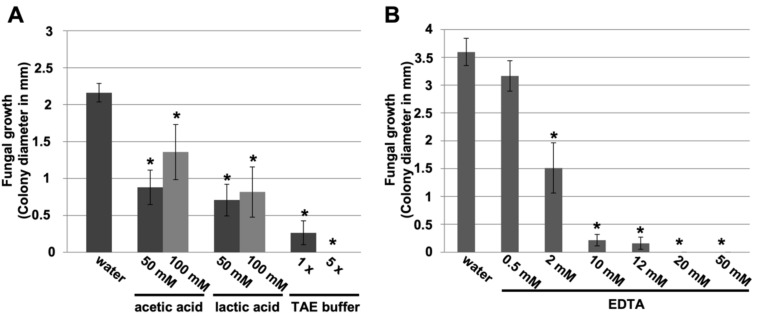
Disease-suppressing effect of carboxylic acids against cucurbit powdery mildew. Melon cotyledon discs were treated with the different compounds and inoculated with *P. xanthii* as described in the Materials and Methods. Powdery mildew symptoms expressed as the *P. xanthii* colony diameter (in mm) were recorded 12 d after pathogen inoculation. (**A**) Initial trials with acetic acid, lactic acid, and TAE (Tris-acetate-EDTA) buffer. (**B**) Trials only with EDTA. Data represent the means of ten samples from two independent experiments, with error bars representing the standard error. The asterisks indicate a statistically significant difference between the treatment and the negative control (water) according to an LSD test (*p* = 0.05). See Appendix A for representative pictures.

**Figure 4 jof-07-01009-f004:**
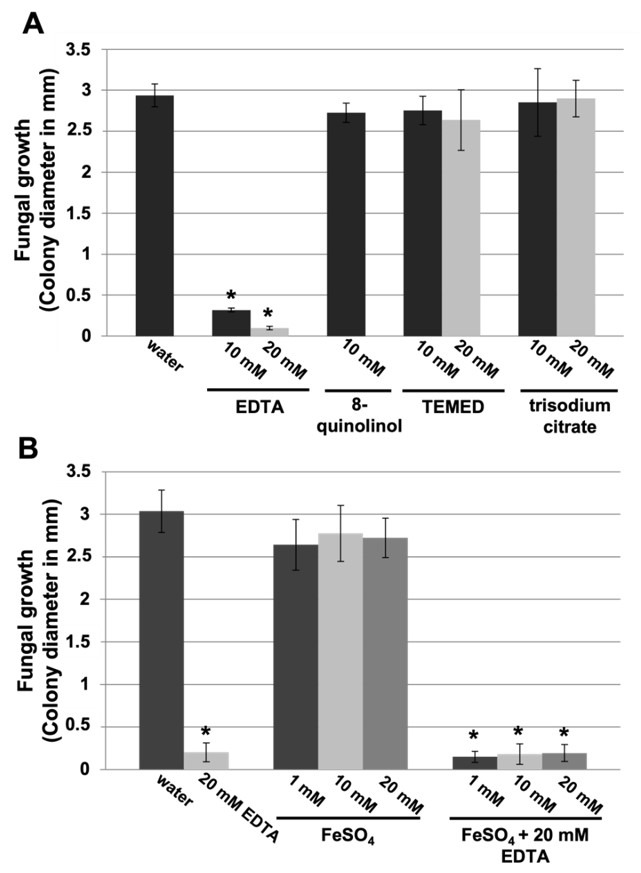
Nondisease-suppressing effect of other chelating agents and disease-suppressing effect of EDTA in presence of metal ions against cucurbit powdery mildew. Melon cotyledon discs were exposed to the different treatments and inoculated with *P. xanthii* as described in the Materials and Methods. Powdery mildew symptoms expressed as the *P. xanthii* colony diameter (in mm) were recorded 12 d after pathogen inoculation. (**A**) Effect of the chelating agents EDTA, 8-quinolinol, TEMED (tetramethylethylenediamine), and trisodium citrate on powdery mildew. (**B**) Effect of ferrous ions (FeSO_4_) on EDTA disease suppression. EDTA was combined with different concentrations of FeSO_4_. Data represent the means of ten samples from two independent experiments, with error bars representing the standard error. The asterisks indicate a statistically significant difference between the treatment and the negative control (water) according to an LSD test (*p* = 0.05). See Appendix A for representative pictures. Data for 20 mM 8-quinolinol are not shown, because it is phytotoxic at this concentration.

**Figure 5 jof-07-01009-f005:**
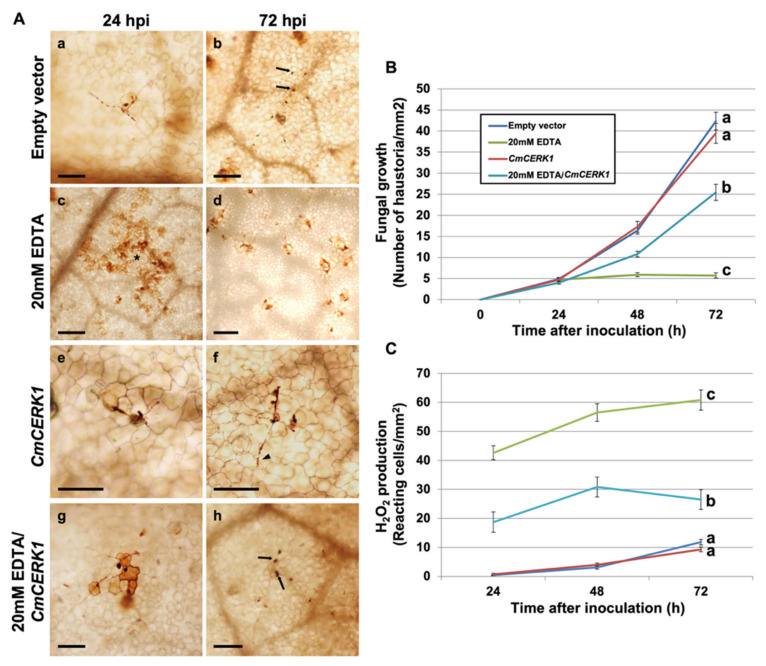
Effect of EDTA on fungal growth and production of hydrogen peroxide. Melon plants were treated with EDTA as described in the Materials and Methods. For RNAi silencing, melon cotyledons were agroinfiltrated, inoculated with *P. xanthii* 24 h after infiltration, and treated with EDTA 24 h after inoculation. Silencing of the melon chitin receptor gene *CmCERK1* was carried out to interfere with host chitin signalling. An empty vector was used as a negative control for RNAi silencing experiments. (**A**) Visualization of fungal structures and H_2_O_2_-producing cells. Detection of hydrogen peroxide was performed by the 3,3′-diaminobenzidine (DAB) uptake method. Pictures were taken at 24 and 72 h after inoculation with *P. xanthii*. Arrowheads indicate *P. xanthii* haustoria; arrows indicate *P. xanthii* hyphae, and asterisks indicate reactive epidermal cells with H_2_O_2_ accumulation. Bars = 50 μm (**e**,**f**), 100 μm (**a**,**b**,**g**,**h**), and 200 μm (**c**,**d**). (**B**) Estimation of fungal growth by haustorial counting. The growth of *P. xanthii* is expressed as the number of haustoria per mm^2^ of transformed tissue. The values are the means of 30 samples from three independent experiments ± standard error. (**C**) Time course analysis of the accumulation of H_2_O_2_ in melon cotyledons. Detection of H_2_O_2_ was performed by the DAB uptake method. The reactive epidermal cells were identified as those containing brown–red precipitates (asterisks). Data represent the number of reactive cells per mm^2^ and are the means of 30 samples from three independent experiments ± standard error. Values with different letters in (**B**,**C**) are significantly different at *p* = 0.05 according to Fisher’s least significant difference test (LSD).

**Figure 6 jof-07-01009-f006:**
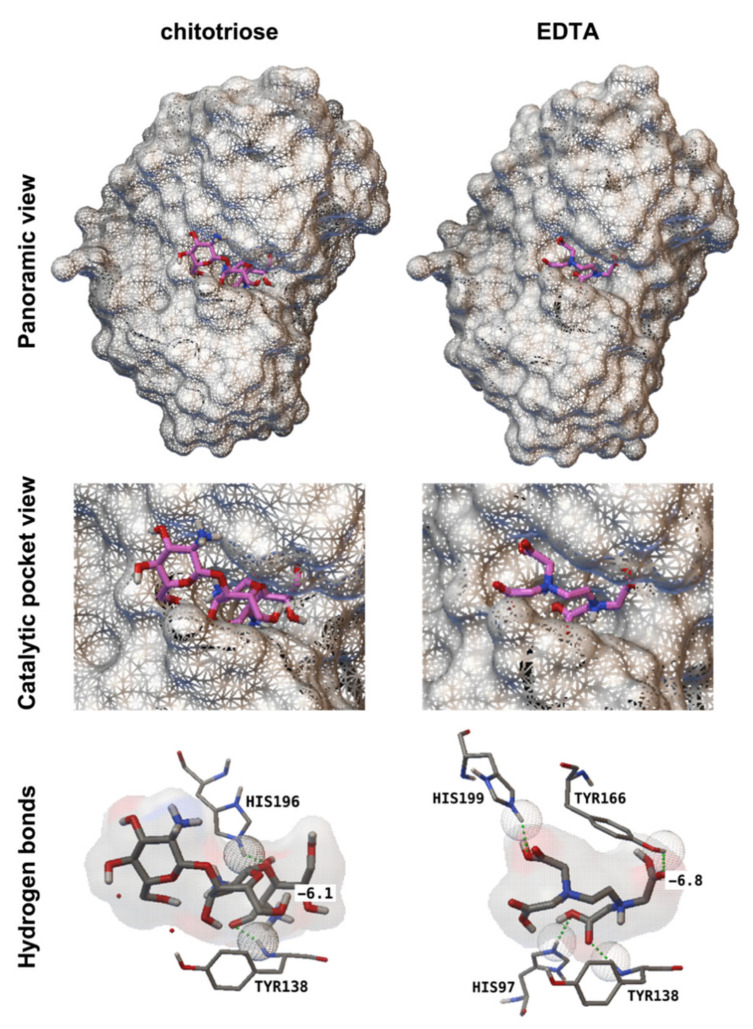
Molecular docking of EDTA and chitotriose to AnCDA. The molecular docking analysis was performed with AutoDock Vina using the 3D model of a CDA protein from *Aspergillus nidulans* (PDB 2y8u) and chitotriose (ChemSpider 108813) and EDTA (ChemSpider 5826) molecules. The results were visualized using AutoDock Tools. Panoramic view (upper images) of CDA protein with the chitotriose and the EDTA molecules docked in the proposed binding site (middle images). The hydrogen bonds (bottom row) between ligands and CDA residues are shown as green dots and wired spheres. EDTA is proposed to bind to AnCDA in the enzyme catalytic pocket (mapped by chitotriose docking and binding site analysis) via four hydrogen bonds that involve His97, Tyr138, Tyr166, and His199. The number indicates the affinity (kcal/mol) of the binding.

**Figure 7 jof-07-01009-f007:**
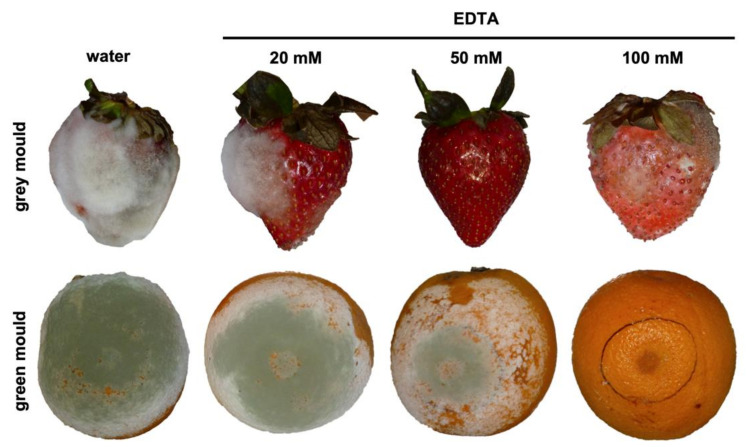
Disease-suppressing effect of EDTA on strawberry grey mould and citrus green mould. The fruits were inoculated with spore suspensions of *B. cinerea* (grey mould) or *P. digitatum* (green mould) as described in Material and Methods and were treated with different EDTA solutions 8 h after pathogen inoculation of the pathogen. Pictures were taken 5 (grey mould) or 6 (green mould) days after inoculation.

**Table 1 jof-07-01009-t001:** Molecular docking analysis of EDTA to a CDA protein from *A. nidulans*.

Compound	Affinity (kcal mol^−1^)	Residues	Hydrogen Bonds (Number)	Distance (Å)
Chitotriose ^1^	−6.1	Tyr138	1	1.73
		His196	1	1.86
EDTA	−6.8	His97	1	1.76
		Tyr138	1	1.80
		Tyr166	1	1.77
		His199	1	1.91

^1^ Molecular docking of chitotriose was also conducted to map the catalytic site.

## Data Availability

The sequence data of the transcripts *PxCDA1* and *PxCDA2* are available in GenBank database under the accession numbers KX495502 and KX495503, respectively.

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
