# Peer review of "Chitin Deacetylase, a Novel Target for the Design of Agricultural Fungicides"

_jof, 2021, doi:10.3390/jof7121009_

Round 1
Reviewer 1 Report
The manuscript entitled Chitin Deacetylase, a Novel Target for the Design of Agricultural Fungicides provides with its approach a possible alternative to current fugicides, impacting on the characteristics of the chitin of the fungal wall. The work allows the identification of a relevant enzyme in the host's immune response as demonstrated by the tests carried out by means of RNAi silencing.
A clear justification for the use of RNAi silencing for the gene deletion instead of the usual knock-out is lacking and what advantages this option provides. Small grammatical errors have to be reviewed.
The discussion lacks studies carried out in other pathogens such as P. digitatum related to chitin syntases that also demonstrate the relevant role in virulence.
The effect of the higher dose of EDTA that results in more infection is not entirely clear.
In any case, the work is very well structured and raises questions that are being resolved little by little. The results provide strength to their conclusions and the hypotheses proposed
Reviewer 2 Report
I made a number of optional suggestions for the manuscript-please see the attachment with the review. This was done to improve the grammar and clarity. Overall I thought this study was very well done.
Line 144-I am confused about the pulverization of the P. xanthii conidia. Or were the cotyledons getting a pulverization treatment with the conidia? Please clarify.
Something looks strange with the water control image in Figure S2A.

Reviewer 3 Report
The authors investigate the chitin deacetylase as a possible new target for fungicide development. The study is very interesting and of high importance. Below are several points that should be addressed before the paper could be published.
Minor corrections are highlighted directly in the pdf file.
The current name of the cucurbit powdery mildew is Podosphaera fusca. Please correct throughout the manuscript (see http://www.indexfungorum.org/Names/NamesRecord.asp?RecordID=464590; https://bsppjournals.onlinelibrary.wiley.com/doi/10.1111/j.1364-3703.2008.00527.x)
In the materials and methods section (2.1 and further) please add the composition of all media used.
In section 2.8 of the materials and methods, please list the compounds and the concentrations you used (it is not sufficient to describe them only in the results).
In section 2.10 of the materials and methods, please add the concentrations of EDTA you used.
Discussion:
you cite two recent papers [37, 38] which describe other mechanisms of suppressing chitin signaling in P. fusca. How would you relate these with CDA activity, as they seem to be partially redundant? How would you explain your results considering also these additional mechanisms of chitin breakdown? You observed complete growth inhibition in the PxCDA silencing line - what about the EWCA and LPMO mediated chitin degradation which should still result in lower activation of chitin-triggered immunity?
How would you explain only a partial restoration of fungal growth after EDTA treatment in CmCERK1 line?
